# Empanelment of health care facilities under Ayushman Bharat Pradhan Mantri Jan Arogya Yojana (AB PM-JAY) in India

**Jaison Joseph** [1] *, **Hari Sankar D.** [1], **Devaki Nambiar** [1,2,3]

1 The George Institute for Global Health, New Delhi, India, 2 Prasanna School of Public Health, Manipal Academy of Higher Education, Manipal, India, 3 Faculty of Medicine, University of New South Wales, Sydney, Australia

* jjoseph@georgeinstitute.org.in, jaison123127@gmail.com.

## Abstract

### Introduction

India's Pradhan Mantri Jan Arogya Yojana (PM-JAY) is the world's largest health assurance scheme providing health cover of 500,000 INR (about USD 6,800) per family per year. It provides financial support for secondary and tertiary care hospitalization expenses to about 500 million of India's poorest households through various insurance models with care delivered by public and private empanelled providers. This study undertook to describe the provider empanelment of PM-JAY, a key element of its functioning and determinant of its impact.

### Methods

We carried out secondary analysis of cross-sectional administrative program data publicly available in PM-JAY portal for 30 Indian states and 06 UTs. We analysed the state wise distribution, type and sector of empanelled hospitals and services offered through PM-JAY scheme across all the states and UTs.

### Results

We found that out of the total facilities empanelled (N = 20,257) under the scheme in 2020, more than half (N = 11,367, 56%) were in the public sector, while 8,157 (40%) facilities were private for profit, and 733 (4%) were private not for profit entities. State wise distribution of hospitals showed that five states (Karnataka (N = 2,996, 14.9%), Gujarat (N = 2,672, 13.3%), Uttar Pradesh (N = 2,627, 13%), Tamil Nadu (N = 2315, 11.5%) and Rajasthan (N = 2,093 facilities, 10.4%) contributed to more than 60% of empanelled PMJAY facilities: We also observed that 40% of facilities were offering between two and five specialties while 14% of empanelled hospitals provided 21–24 specialties.

### Conclusion

A majority of the hospital empanelled under the scheme are in states with previous experience of implementing publicly funded health insurance schemes, with the exception of Uttar

**Data Availability Statement:** All relevant data are within the manuscript and supporting information files.

**Funding:** We wish to indicate that this work was supported by the Wellcome Trust/DBT India Alliance Fellowship (https://www.indiaalliance.org) Grant number IA/CPHI/16/1/502653) awarded to Dr. Devaki Nambiar. The funder had no role in study design, data collection and analysis, decision to publish, or preparation of the manuscript. The funder provided support in the form of salaries and research materials and field work support for authors DN, HS and JJ but did not have any additional role in the study design, data collection and analysis, decision to publish, or preparation of the manuscript. The specific roles of these authors are articulated in the 'author contributions' section.

**Competing interests:** We are in full adherence of Plos One standard on sharing data and materials. I wish to confirm that all authors have no competing interests to declare.

**Abbreviations:** AB PM-JAY, Ayushman Bharat Pradhan Mantri Jan Arogya Yojana; BPL, Below the Poverty Line; CGHS, Central Government Health Scheme; CHC, Community Health Centres; COVID19, Novel Coronavirus SARS-CoV-2; DALYs, Disability Adjusted Life Years; EHCF, Empanelled Health Care Facilities; ESI, Employee State Insurance; GDP, Gross Domestic Product; HEM, Hospital Empanelment Module; IPHS, Indian Public Health Standards; NHA, National Health Authority; NHM, National Health Mission; OPD, Outpatient Department; PFHIS, Publicly Funded Health Insurance Schemes; PHC, Primary Health Centres; PSU, Public Sector Undertaking; RSBY, Rashtriya Swasthya Bima Yojana; SDG, Sustainable Development Goal; SDH, Sub District Hospital; SECC, Socio-Economic Caste Census; SHA, State Health Agencies; THE, Total Health Expenditure; UHC, Universal Health Coverage; USD, United States Dollars; UTs, Union Territories.

Pradesh. Reasons underlying these patterns of empanelment as well as the impact of empanelment on service access, utilisation, population health and financial risk protection warrant further study. While the inclusion and regulation of the private sector is a goal that may be served by empanelment, the role of public sector remains critical, particularly in underserved areas of India.

# Introduction

Sustainable Development Goal (SDG) 3.8 seeks to ensure the health and wellbeing of all by achieving Universal Health Coverage (UHC) [1]. UHC emphasises on the importance of equity in access to quality health care for everybody without risking financial hardship [2]. India faces enormous challenges in moving towards UHC, which include suboptimal access, insufficient availability of services, poor quality health service delivery, and high out of pocket expenditure [3]. In 2015, global spending on health was USD 10 trillion and total health spending is projected to double to USD 20 trillion (18 trillion to 22 trillion) in 2040 [4]. India and China have increased the pooled per capita health spending by more than 265% between 1995–2015, a positive step in the direction of UHC [4]. However, India's financial allocation to health sector remains inadequate. The country's 2019 National Health Accounts reported Total Health Expenditure (THE) of Rs. 581,023 ten million (USD 767.7 trillion) which is 3.8% of GDP and Rs.4,381 (USD 58) per capita for the year 2016–17. Out of pocket expenditure is a major contributor to THE (at 58.7%) [5]. Analysis of catastrophic health expenditure trends in India show an increasing trend in last two decade with households with older people suffering the most [6]. There is evidence to suggest this will only increase in the context of the coronavirus pandemic [7]. High level of fragmentation in the sources of revenues and low risk pooling mechanisms in the country resulted in high out of pocket expenditure (about 62% of expenditure coming directly from households) especially among the poor and near poor [8].

However, the national health policy reflects the commitment towards achieving UHC through developing institutional mechanisms to improve the coverage and access to health services. The National Health Policy of 2017 expressed a commitment to increase the government health expenditure from 1.15% to 2.5% of the GDP by 2025 [9]. A flagship effort in this direction was the Ayushman Bharat Program [10], launched in 2018 to holistically address the primary, secondary and tertiary level health needs of the population by ensuring continuum of care [11]. The two interrelated components of Ayushman Bharat are: 1) Health and Wellness Centres (HWCs) to provide comprehensive primary care services and 2) the Pradhan Mantri Jan Arogya Yojana (PM-JAY) to provide secondary and tertiary care services which will enable the realization of the aspiration for UHC [12]. HWCs are upgraded primary care facilities intended to progressively expand access to comprehensive primary health care, free essentials drugs as well as diagnostics services; whereas the PM-JAY aims to provide financial protection for secondary and tertiary care to bottom 40% of India's population [12]. PM-JAY aims to ensure improved access to good quality healthcare services through a combination of public and private empanelled providers for everyone without financial hardship [13].

PM-JAY has evolved with learnings from longstanding Indian Publicly Funded Health Insurance Schemes (PFHIS) for formal sector like Employee State Insurance (ESI,1952) and Central Government Health Scheme (CGHS,1954) in the formal sector, and the Rashtriya Swasthya Bima Yojana (RSBY, 2008) for the informal sector. However, the benefits and coverage offered under PM-JAY are much larger than these schemes [13]. AB PM-JAY is designed

to meet the hospitalisation expense in cashless mode with a coverage of Rs.5,00,000/- (approx. $ 6,800) per family per annum to entitled beneficiaries on a floater basis i.e. the total insured amount can be used by one or all the members of the family (see characteristics of the scheme in Table 1).

It is important to note that since health is a state subject in India, the implementation model of PM-JAY varies across the country, and employs the concept of cooperative federalism, where components of program design, implementation and funding across federal and state levels are shaped in part by flexibilities offered by the scheme but also state context and prior experience with implementing public insurance (see Table 1). There is a strong possibility that these design variations may have a differential impact while operationalizing, which is the starting point for our analysis apart from addressing concerns that the core design of PM-JAY itself may be inadequate to fulfil the requirements of financial risk protection envisioned through UHC [19–22].

We place emphasis in this analysis on scheme empanelment at the state level, which is one of the four core functions of insurance (alongside enrolment, claims processing and grievance redressal). Under PM-JAY, empanelment of hospitals is processed through an online IT platform called the Hospital Empanelment Module (HEM). Based on defined criteria (see "who provides services" in Table 1) [23], the decision on empanelment of hospitals is subject to approvals from the empanelment committees at the district and state/union territory levels. Selection of providers is critical in strategic purchase of care from a mix of public and private sector and is often advocated to ensure competition and increase quality of delivery [24]. This option may not always be available as India's private sector health providers are mostly

**Table 1. Key features of the Pradhan Mantri Jan Arogya Yojana (PMJAY).**

| | |
|---|---|
| Purchasers | National Health Authority (NHA) through State Health Agencies (SHA) with flexibility for states to implement the scheme and purchase care through one of three modes: a public trust, a third-party Insurance, or a combination (i.e. trust and Insurance) |
| What services are purchased? | a) Health insurance coverage of Rs. 5,00,000 (roughly $ 6,800) per family annually for secondary and tertiary care hospitalization.<br>b) Covering 3 days of pre-hospitalization and post hospitalization charges up to 15 days<br>c) As of 2019, services comprise of nearly 1,393 procedures (1,083 are surgical, 309 medical and 1 unspecified package) covering all the costs incurring for treatment, drugs and consumables diagnostics, and various user fees [14]. However, states are given power to restrict certain treatment packages for public sector only<br>d) There is no restriction on family size, age or gender and beneficiaries can avail cashless treatment from an empanelled healthcare provider |
| Who uses the services? | Enrolled Population falling under the category<br>• Below the Poverty Line (BPL) in the Socio-Economic Caste Census (SECC)<br>• Existing Rashtriya Swasthya Bima Yojana (RSBY) beneficiaries<br>• State notified categories |
| Who provides services? | **Public-** All public hospitals (including ESIC [15]) equipped with inpatient facilities (Community Health Centre level and above) are empanelled by default [16].<br>**Private and not for profit hospitals–**Hospitals meeting the minimum criteria established by National Health Authority (NHA) which include qualified doctor and nurse presence, in-patient beds with staff, medical and surgical service availability (including human resources around the clock, support systems, ambulance facilities 24x 7 with technically qualified staff) [16]. |
| How are providers paid? | Based on the treatment package, public and private hospitals have the same package rate, which may be specified, like a surgical package for which there is case based bundled payment or unspecified, for which a claimant will negotiate with pre-approval by intermediary/ SHA. |

Source: Categories based on Etiba & colleagues [17]; data from PM-JAY public websites/portal [11, 18], compiled by authors

urban-centric and the empanelment of private facilities under PM-JAY scheme varies from state to state [24]. For example, in Bihar half of all private providers in the state are situated in the capital city Patna and out of the 38 districts in the state 14 districts do not have a single private provider registered in the scheme [24]. Similar studies on RSBY have also reported that empanelled private hospitals tend to be greater in urban areas, providing a narrow and selective range of packages/conditions which were profitable, and only a fixed number of beds were earmarked for RSBY patients [25]. A study in Chhattisgarh on examining the availability of empanelled hospitals reported poor availability of private hospital services in geographically challenged areas [26]. Studies show that availability of hospital care had increased for RSBY enrolees [27], but access has remained skewed due to absence of empanelled hospitals in many geographically challenged areas, leading to non-utilization of services [28].

One of the key objectives of the PM-JAY is to increase the availability and choice of healthcare facilities such that beneficiaries can avail free treatment, through public or private healthcare providers [29]. PM-JAY offers portability of care to its beneficiaries which is essential for India's vast geography and high interstate migration of workforce. The approach to empanelment of private providers by different states is governed by their existing public health infrastructure as well as state capacity to provide treatment for different specialties [29]. Certain states have adopted a policy to empanel only a limited number of hospitals that meet their requirement (e.g. Maharashtra), while certain States have reserved packages for public hospitals (e.g. Bihar, Madhya Pradesh) and states such as Uttarakhand have adopted a policy of referral from a public hospital for every procedure to be carried out at a private hospital [29]. Some abuse prone packages like hysterectomy have been reserved exclusively for public hospitals. Some states like Kerala, Jharkhand, Madhya Pradesh etc have adopted the list as is, while others like Karnataka, Tamil Nadu, Gujrat etc have added more packages depending on capacity of public hospitals and other local factors [29]. In most of the tertiary care schemes, the minimum criteria for empanelment is 50 beds, however in order to secure a more geographical accessible network, hospitals with 10 beds or more are also empanelled under PM-JAY [30]. It is estimated that on an average only a single bed is available for 1,844 persons in public hospitals and nine beds per 10,000 population in India against the global average of 30, in which private sector accounts for nearly half (49%) of the beds available [31].

While there are a fair number of state specific analyses, very few studies have explored the availability and distribution of EHCFs across India under public health insurance. With the UHC goal of equitable access in mind, we looked at the building blocks of insurance, specifically the first component of empanelment and sought to fill some gaps in knowledge. Specifically, we were interested in a) where people could access care under PM-JAY, and b) what services were being provided in these hospitals. Given the evidence, we sought to understand the public/private mix of hospital empanelment while also exploring patterns related to models of implementation, which vary across states.

## Methodology

### Study design

Secondary analysis of cross-sectional data using descriptive statistical methods was conducted to determine the geographical distribution, type and sector of empanelled hospitals and services offered through PM-JAY scheme. Data from all the states and UTs were taken into consideration for this study based on their mode of implementation. The study did not use any personal identifiable information and the data used for the study is available in the public domain (our extracted dataset is included as S1 File). Thus, this study was exempt from review by an ethics committee.

### Data sources

This is a descriptive analysis of data sourced for the reference period February to March 2020. The analysis was carried out using data accessed from the Government of India's PM-JAY website on the public and private hospitals empanelled under the scheme across Indian 30 states and 06 Union Territories (UTs) [32]. Information available included: a) hospital name b) hospital type (public, private for profit, private not for profit) c) hospital address d) hospital email e) hospital contact f) specialties empanelled and g) specialties upgraded. We prepared a database of all hospitals enlisted under the scheme with above mentioned variables in Microsoft Excel [33], for analysis. Further we obtained information on the state wise number of eligible household's from "state at a glance" [34], on the PM-JAY website and estimated the number of eligible beneficiaries manually using the average family size from Indian Socio Economic Caste Census Report 2011 [35]. The analysis was done by classifying states based on the mode implementation as hybrid mode, insurance mode, trust mode, or the National Health Claims Platform (NHCP), based on information obtained from PM-JAY website. Information on specialties offered in 15,177 hospitals for which information were provides as indicated in the PMJAY website was recoded and linked for analysis in Microsoft Excel and the state wise count of facilities providing specialties under each sector was obtained.

Public sector facilities were classified based on the Indian Public Health Standards (IPHS) as Sub-centres, Primary Health Centres (PHCs), Community Health Centres (CHCs), Sub-District and District Hospitals [36]. Other type of facilities like medical colleges were categorized as given, while facilities like Public Sector Undertaking (PSU) hospitals, Railway and Military hospitals were grouped as "other" facilities. For understanding the availability of beds in public health facilities, sanctioned beds were collated from reports available on state health department and National Health Mission (NHM) websites wherever available else the maximum bed strength as per IPHS standard was allocated and bed strength was estimated.

As the data are dynamic, there were variations in total number of empanelled hospitals under the scheme ranging from 18,699 on February 2020 to 20,257 on March 2020. This study focuses on the 20,257 hospitals empanelled as on March 2020 to analyse the spread and access to network hospitals under the scheme. We conceptualized the empanelment data of facilities through an access framework by looking at the data through state wise distribution of facilities, sector wise (public/private), by mode of implementation, and availability of specialties offered through EHCFs.

### Results

We analysed the overall distribution of EHCF by public and private sector and by the mode of implementation adopted by the states viz. trust mode, hybrid mode and insurance mode. Out of the total facilities empanelled (N = 20,257) under the scheme, more than half 11,367 (56%) were in the public sector, 8,157 (40%) facilities were private for profit, and 733 (4%) were private not for profit entities. Even though there was not much difference in distribution of empanelled healthcare facilities by sector across the different modes of implementation, the trust model dominates (60.4% of empanelled facilities) followed by hybrid model 30.9% and with insurance mode accounting for 7.7% of empanelled hospitals (see Table 2).

It was noted that 8% of the EHCFs in the public sector were empanelled based on bed occupancy parameter and belonged to Ministry of Home Affairs (562), Ministry of Railways (91), Ministry of Power (52), Ministry of Coal (42), Institute of National Excellence (14), Ministry of Heavy Industries and Public Enterprises (14), Ministry of Labour and Employment (ESIC) (14), Ministry of Steel (13), Ministry of Shipping (7), Ministry of Défense (7), Ministry of Mines (3), Ministry of Petroleum and Natural Gas (2) and New Delhi Municipal Council (1).

**Table 2. Overall distribution of PMJAY empanelment by mode of implementation and sector.**

| Mode of Implementation | Public (row percentages) | Private for Profit (row percentages) | Private Not for Profit (row percentages) | Total EHCFs (column percentages) |
|---|---|---|---|---|
| Trust Mode | 6988 (57%) | 4796 (39%) | 454 (4%) | **12238 (60.4%)** |
| Hybrid Mode | 3419 (55%) | 2695 (43%) | 158 (3%) | **6272 (30.9%)** |
| Insurance Mode | 811 (52%) | 644 (41%) | 115 (7%) | **1570 (7.7%)** |

Note

[a] Four states are not implementing PMJAY scheme, however public sector facilities are empanelled in these states for ensuring portability accounting for 153 facilities (Delhi 53, Odisha 29, Telangana 12 and West Bengal 59)

[b] 24 Hospitals are empanelled under National Health Claims Platform (NHCP) which include hospitals empanelled by National Health Authority (NHA) at National level.

Source: Data from PM-JAY portal [16], compiled by authors

State wise distribution of hospitals (see Table 3) showed that five states contributed to more than 60% of empanelled facilities under PMJAY: Karnataka with 2,996 (14.9%), Gujarat with 2,672 (13.3%), Uttar Pradesh with 2,627 (13%), Tamil Nadu with 2315 (11.5%) and Rajasthan with 2,093 facilities (10.4%). Among these states, Rajasthan, Uttar Pradesh and Karnataka implement the scheme through a trust model and the remaining through hybrid models. Karnataka and Gujarat had relatively high proportions of public facility empanelment as a share of total empanelment, while Rajasthan had a large share of private (for profit) empanelment, and Tamil Nadu and Uttar Pradesh had empanelment shares close to being split across public and private sectors.

We also found that nine states and one union territory (Andhra Pradesh, Goa, Haryana, Jharkhand, Kerala, Maharashtra, Rajasthan, Punjab and Uttar Pradesh, and Chandigarh) had a higher proportion of private facilities (for profit and not for profit combined) empanelled when compared with the public sector facilities. Otherwise, the public sector dominated empanelment across states.

Given the dominance of the public sector at the national level, we sought to understand empanelment in this category further (see S1 Table). Results show that out of the total public hospitals empanelled under the scheme, 40.4% were primary health care facilities and 29.1% were secondary care facilities (comprising Community Health Centres (CHC)/Urban Community Health Centres (UCHC) and Sub District Hospitals (SDH)). District hospitals accounted for 8.4% and medical colleges offering tertiary care accounted for 7.3% of facilities. The number of tertiary care facilities was fewer than the other primary and secondary facilities in the sector.

State wise distribution of hospitals (see S2 and S3 Tables) showed that out of the total public EHCFs, 39% of facilities fell under the category of Primary Health Centres and in five states (Gujarat 79%, Meghalaya 68%, Bihar 55%, Karnataka 82%, Mizoram 66%) more than 50% of empanelled facilities were PHCs. Further, 30% of facilities were Community Health Centres which deliver some specialty services and in nine states CHCs contribute to more 50% of empanelled facilities. This pattern of higher proportion of PHC empanelment is visible in trust (43%) and hybrid modes (42%).

State wise distribution of estimated number of beds in the public sector empanelled hospitals (see S4 and S5 Tables) showed that 75% of the bed availability was in secondary and tertiary care facilities. It is to be noted states like Delhi, Odisha, Telangana and West Bengal, which do not implement the scheme, also offer significant number of tertiary care beds under the scheme, ensuring portability. Uttar Pradesh had the highest absolute number of beds in

**Table 3. State wise distribution of PMJAY empanelment by mode of implementation and sector.**

| State Name | Mode of Implementation | Public | Private (For Profit) | Private (Not for Profit) | Total |
|---|---|---|---|---|---|
| Gujarat | Hybrid | 1817 (68.0%) | 765 (28.6%) | 90 (3.4%) | **2672** |
| Jharkhand | Hybrid | 274 (36.1%) | 425 (56.1%) | 59 (7.8%) | 758 |
| Maharashtra | Hybrid | 123 (23.3%) | 404 (76.7%) | 0 | 527 |
| Tamil Nadu## | Hybrid | 1205 (52.1%) | 1101 (47.6%) | 9 (0.4%) | **2315** |
| Dadra & Nagar Haveli | Insurance | 4 (100.0%) | 0 | 0 | 4 |
| Daman & Diu | Insurance | 3 (100.0%) | 0 | 0 | 3 |
| Jammu & Kashmir | Insurance | 126 (79.2%) | 26 (16.4%) | 7 (4.4%) | 159 |
| Kerala | Insurance | 187 (46.6%) | 182 (45.4%) | 32 (8.0%) | 401 |
| Meghalaya | Insurance | 163 (92.1%) | 9 (5.1%) | 5 (2.8%) | 177 |
| Nagaland | Insurance | 75 (92.6%) | 6 (7.4%) | 0 | 81 |
| Puducherry | Insurance | 12 (60.0%) | 6 (30.0%) | 2 (10.0%) | 20 |
| Punjab | Insurance | 241 (33.2%) | 415 (57.2%) | 69 (9.5%) | 725 |
| Andhra Pradesh | Trust | 225 (32.1%) | 477 (67.9%) | 0 | 702 |
| Andaman & Nicobar | Trust | 3 (100.0%) | 0 | 0 | 3 |
| Arunachal Pradesh | Trust | 5 (100.0%) | 0 | 0 | 5 |
| Assam | Trust | 160 (53.9%) | 117 (39.4%) | 20 (6.7%) | 297 |
| Bihar | Trust | 571 (71.3%) | 196 (24.5%) | 34 (4.2%) | 801 |
| Chandigarh | Trust | 5 (27.8%) | 11 (61.1%) | 2 (11.1%) | 18 |
| Chhattisgarh | Trust | 714 (75.9%) | 227 (24.1%) | 0 | 941 |
| Goa | Trust | 11 (45.8%) | 13 (54.2%) | 0 | 24 |
| Haryana | Trust | 166 (30.9%) | 323 (60.1%) | 48 (8.9%) | 537 |
| Himachal Pradesh | Trust | 143 (70.1%) | 52 (25.5%) | 9 (4.4%) | 204 |
| Karnataka | Trust | 2517 (84.0%) | 475 (15.9%) | 4 (0.1%) | **2996** |
| Lakshadweep | Trust | 1 (100.0%) | (0.0%) | 0 | 1 |
| Madhya Pradesh | Trust | 411 (77.8%) | 92 (17.4%) | 25 (4.7%) | 528 |
| Manipur | Trust | 50 (87.7%) | 7 (12.3%) | 0 | 57 |
| Mizoram | Trust | 86 (89.6%) | 10 (10.4%) | 0 | 96 |
| Rajasthan | Trust | 595 (28.4%) | 1498 (71.6%) | 0 | **2093** |
| Sikkim | Trust | 9 (90.0%) | 0 | 1 (10.0%) | 10 |
| Tripura | Trust | 100 (98.0%) | 2 (2.0%) | 0 | 102 |
| Uttarakhand | Trust | 123 (62.8%) | 51 (26.0%) | 22 (11.2%) | 196 |
| Uttar Pradesh | Trust | 1093 (41.6%) | 1245 (47.4%) | 289 (11.0%) | **2627** |
| **Total** | | **11218 (56%)** | **8135 (41%)** | **727 (4%)** | **20080** |

**Note**: 177 EHCFs accounting for states not implementing the scheme and empanelled by NHCP not included for analysis.

## Tamil Nadu employs its own information system for empanelment which considers individual departments in individual facilities. Therefore, the total number of 'empanelled' facilities is likely lower than what is reported above.

Source: Data from PM-JAY portal [16], compiled by authors

public sector (12%) but the availability of bed per hundred thousand eligible population was only 87 (see S6 Table). Sikkim and Goa had the highest beds available per 100,000 population.

At the national level, AB-PMJAY has empanelled 3 hospitals per 100,000 eligible population, 2 in public and 1 in private sector, respectively (see Table 4). In Chandigarh, the rate of private hospitals empanelled per 100,000 population was double that in the public sector, in Rajasthan it was 3 times greater in private than in public, whereas in Nagaland it was 6 times higher in public than in private. In Tamil Nadu, Kerala and Assam we saw roughly matched public and private sector empanelment ratio to population.

**Table 4. Empanelment of hospitals across the states relative to population, by model of implementation.**

| State/UT Name | Hospitals / 100,000 eligible beneficiaries | Public Hospitals / 100,000 eligible beneficiaries | Private Hospital / 100,000 eligible beneficiaries |
|---|---|---|---|
| **Hybrid** | | | |
| Gujarat | 7 | 5 | 2 |
| Jharkhand | 2 | 1 | 2 |
| Maharashtra | 1 | 0 | 1 |
| Tamil Nadu | 4 | 2 | 2 |
| **Insurance** | | | |
| DNH and DD | 1 | 1 | 0 |
| Jammu & Kashmir | 5 | 4 | 1 |
| Kerala | 2 | 1 | 1 |
| Meghalaya | 4 | 4 | 0 |
| Nagaland | 7 | 6 | 1 |
| Puducherry | 5 | 3 | 2 |
| Punjab | 3 | 1 | 2 |
| **Trust** | | | |
| Andaman & Nicobar | 3 | 3 | 0 |
| Andhra Pradesh | 1 | 0 | 1 |
| Arunachal Pradesh | 0 | 0 | 0 |
| Assam | 2 | 1 | 1 |
| Bihar | 1 | 1 | 0 |
| Chandigarh | 6 | 2 | 4 |
| Chhattisgarh | 4 | 3 | 1 |
| Goa | 15 | 7 | 8 |
| Haryana | 6 | 2 | 4 |
| Himachal Pradesh | 9 | 6 | 3 |
| Karnataka | 5 | 5 | 1 |
| Lakshadweep | 12 | 12 | 0 |
| Madhya Pradesh | 1 | 1 | 0 |
| Manipur | 4 | 3 | 0 |
| Mizoram | 10 | 9 | 1 |
| Rajasthan | 4 | 1 | 3 |
| Sikkim | 5 | 5 | 1 |
| Tripura | 5 | 5 | 0 |
| Uttar Pradesh | 3 | 1 | 2 |
| Uttarakhand | 3 | 2 | 1 |
| **Total** | 3 | 2 | 1 |

The distribution of specialty care packages available in public and private empanelled hospitals are depicted in Fig 1. It is observed that 40% of facilities were offering 2–5 specialty care and 14% of empanelled hospitals provided 21–24 specialties.

Across facilities, the key specialties (Fig 2) available in the empanelled hospitals were general medicine (74.4%), emergency room packages (71.8%), general surgery (62.1%), obstetrics and gynaecology (58.9%), and orthopaedics (50%).

Tertiary care packages were offered mostly in public sector hospitals (see Table 5) Super-specialty packages like cardiothoracic surgery (79.6%), medical oncology (73.9%), cardiology (65%), were available in most of the empanelled tertiary care public hospitals, while their share in private sector were reported lower. The private sector had greater relative provision of orthopaedics (56.3%) general surgery (55.2%), urology services (51.5%) and obstetrics & gynaecology (49.5%).

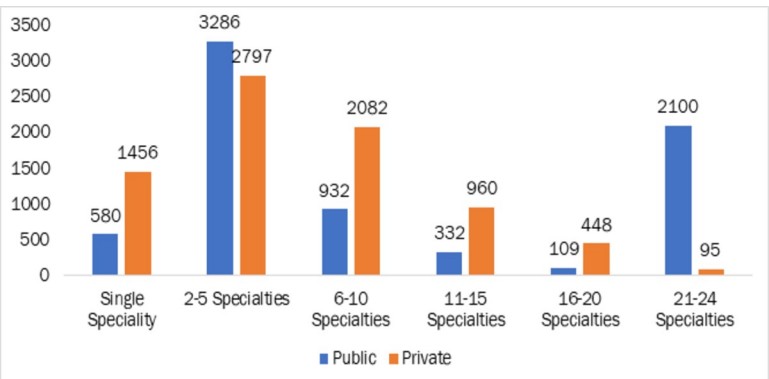

**Fig 1. Number of empanelled hospitals providing specialty care as per PMJAY Package.** Note: The data of 15177 EHCFs with available specialty information from PMJAY website was used for this analysis Source: Data from PM-JAY portal [16], compiled by authors.

## Discussion

The availability of timely, robust data in the public domain is the cornerstone of any monitoring process and is key to the transparency, accountability of any health system. The AB—PMJAY website offers detailed facility-related information which can provide useful insights regarding the nuances of scheme functioning and progress. The present study looked at data on PMJAY empanelment of facilities to understand the nature of distribution of services to its target population.

EHCFs are an essential element for any health insurance programme: identifying and empanelling providers working towards both quality and access [37]. Our study found that more than half of the EHCFs were government facilities. This was consistent with an earlier study on public health insurance which reported that majority of the hospitals empanelled under PFHIs are from public sector and in low income states of the country, empanelled private hospitals were concentrated in a few pockets, had low willingness to participate, which authors argued would limit access to healthcare for intended beneficiaries [38]. Results from a recent study in India's Aspirational Districts [39], showed that 9 states had no private hospitals empanelled in any aspirational districts and the share of hospitals available to provide key tertiary care services in aspirational districts were less when compared with other districts [40].

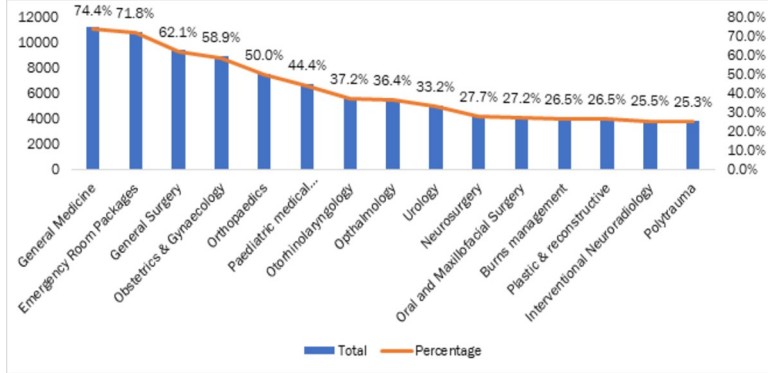

**Fig 2. Top 15 specialties available across the states as per PMJAY package.** The data of 15177 EHCFs with available specialty information from PMJAY website was used for this analysis Source: Data from PM-JAY portal [16], compiled by authors.

**Table 5. Services available through empanelled facilities.**

| Specialties | Total Available Empanelled Hospitals (N = 15177) | Availability in Private Empanelled Hospitals | Availability in Public Empanelled Hospitals |
|---|---|---|---|
| General Medicine | 11295 (74.4%) | 4950 (43.8%) | 6345 (56.2%) |
| Emergency Room Packages | 10894 (71.8%) | 4634 (42.5%) | 6260 (57.5%) |
| General Surgery | 9425 (62.1%) | 5201 (55.2%) | 4224 (44.8%) |
| Obstetrics & Gynaecology | 8938 (58.9%) | 4421 (49.5%) | 4517 (50.5%) |
| Orthopaedics | 7582 (50.0%) | 4267 (56.3%) | 3315 (43.7%) |
| Paediatric medical management | 6731 (44.4%) | 2963 (44.0%) | 3768 (56.0%) |
| Otorhinolaryngology | 5648 (37.2%) | 2417 (42.8%) | 3231 (57.2%) |
| Ophthalmology | 5525 (36.4%) | 2329 (42.2%) | 3196 (57.8%) |
| Urology | 5035 (33.2%) | 2595 (51.5%) | 2440 (48.5%) |
| Neurosurgery | 4203 (27.7%) | 1864 (44.3%) | 2339 (55.7%) |
| Oral and Maxillofacial Surgery | 4135 (27.2%) | 1028 (24.9%) | 3107 (75.1%) |
| Burns management | 4022 (26.5%) | 1561 (38.8%) | 2461 (61.2%) |
| Plastic & reconstructive | 4022 (26.5%) | 1561 (38.8%) | 2461 (61.2%) |
| Interventional Neuroradiology | 3876 (25.5%) | 1621 (41.8%) | 2255 (58.2%) |
| Polytrauma | 3841 (25.3%) | 1391 (36.2%) | 2450 (63.8%) |
| Cardiology | 3637 (24.0%) | 1272 (35.0%) | 2365 (65.0%) |
| Paediatric surgery | 3522 (23.2%) | 1152 (32.7%) | 2370 (67.3%) |
| Neo-natal | 3368 (22.2%) | 904 (26.8%) | 2464 (73.2%) |
| Surgical Oncology | 3122 (20.6%) | 873 (28.0%) | 2249 (72.0%) |
| Medical Oncology | 3065 (20.2%) | 800 (26.1%) | 2265 (73.9%) |
| Cardio-thoracic & Vascular surgery | 2728 (18.0%) | 556 (20.4%) | 2172 (79.6%) |
| Mental Disorders Packages | 2635 (17.4%) | 204 (7.7%) | 2431 (92.3%) |
| Paediatric cancer | 2626 (17.3%) | 265 (10.1%) | 2361 (89.9%) |
| Radiation Oncology | 2514 (16.6%) | 278 (11.1%) | 2236 (88.9%) |

Source: Data from PM-JAY portal [16], compiled by authors

This is also discussed in the PMJAY policy brief which reiterates the low participation of private sector in aspirational districts of the country [41]. The skewed distribution of private hospitals in states with low per capita income is an area of concern as a significant proportion of the eligible population under AB-PMJAY is concentrated in these states [41, 42]. A report on a government funded health insurance scheme in Maharashtra reported that the unwillingness of the multi-specialty private hospitals to participate in the scheme negatively affected the availability of services; infrastructural lacunae in the rural government hospitals continued unaddressed [43]. PMJAY was intended to increase access to services for the poor; yet the distribution of empanelled hospitals suggests poor service availability in many states. Participation of the private sector in public health insurance depends on profitability; thus, low premia and/or price-controlled package rates may discourage participation [44, 45]. This is clearly an area for further research.

PMJAY allows beneficiaries to access healthcare services free of cost through empanelled facilities anywhere in the country. Our analysis showed that states where the scheme was not implemented (like Delhi, Telangana, Odisha and West Bengal), accounted for 28% of bed share of public empanelled facilities. The national interoperability is a unique feature which is showcased as an asset of the scheme. India has huge interstate migration of labourers–this feature is therefore vital in principle. A survey done among the patients who have utilized the

portability feature under PM-JAY from other state/districts revealed that lack of required services in their home state is by far the most common reason for seeking cross-border care [46]. Particularly in a COVID19 context, portability will be increasingly important and must continue to be assessed using deeper analytic approaches.

Access to health services in India is highly inequitable, with major disparities in health outcomes across income, gender, tribe, caste, and geographically defined population subgroups [47]. Our study found that Rajasthan, Uttar Pradesh, Gujarat, Karnataka and Tamil Nadu together accounted for more than 50% of total hospitals empanelled under the scheme. The ratio of empanelled hospitals to population was 4 hospitals per hundred thousand population in these five states as compared to 2 hospitals per hundred thousand population for all remaining 25 states/UTs combined, even as the latter account for an estimated share of 55% of eligible beneficiaries under the scheme. This is consistent with previous study which reported inequitable distribution of empanelled hospitals, especially of private hospitals, within six Indian states [48]. All the states with high empanelment are implementing the scheme under trust mode wherein the autonomy of empanelment relies with the State Health Agency (SHA). All the above-mentioned states except for Uttar Pradesh have over a decade of experience in successfully implementing Publicly Funded Health Insurance Schemes and have governance and institutional structures that likely facilitated implementation of PMJAY. Features of the health system and the broader political economy of empanelment warrants further study, however, as it is likely that system actors also have played a role in this pattern. Health systems research methods that explore governance may offer insights in this regard [49, 50].

The distribution of empanelment (private versus public) does not seem to vary substantially by mode of implementation (trust, insurance agency, or hybrid model) as of now. Among the public hospitals empanelled under the scheme by various modes, the insurance mode had more secondary and tertiary care facilities than other two. The model as it stands right now is similar to Indonesia's Jamkesmas provider network, even as in this case, the network has not significantly increased benefit package availability in remote, rural locations [51]. Evidence from India suggests that neither trust nor insurance company purchasing models are associated with increased utilisation of hospital care in southern Indian states (where these programmes have some maturity), nor is there an association with out of pocket expenditure associated with enrolment [52]. Another study has noted that hospital insurance reforms in LMICs like Kenya require particular attention to design, where threats like "purchaser capture" may prove unsustainable [53].

Results showed that primary care facilities accounted for 25% of the total bed share under PM-JAY. These facilities are officially available to the entire population technically free or for nominal charges [54]. The National Sample Survey 2018 defines PHCs as institutions that provide curative OPD services, ante natal check-ups and deliveries (4–6 beds to conduct delivery) with limited facilities for in-patient treatment [55]. Given the large evidence base suggesting that public facilities have poor infrastructure, under-staffing and lack of equipment and medicines [56–58], it is unclear how PMJAY empanelment relates to the service design at the public primary care level. On the one hand, empanelment under PMJAY is based on a set of criteria for all facilities and may have contributed towards upgradation of primary care facilities, although it is unlikely that this upgradation may have occurred uniformly at the scale and pace of empanelment. On the other, pressure to maximise empanelment numbers may have resulted in relaxation of empanelment criteria in the public sector. There is lack of evidence on these processes and a strong need for further research.

More fundamentally moreover, the precise implication of empanelment in public facilities is unclear; it is suggested that while empanelment may change payment mechanisms on the provider side, there is likely little difference for patients, who remain entitled to free or

subsidized care at the government hospitals in any case [59]. It remains to be seen whether empanelment is a pathway to improvements in facility quality across public and private sectors and whether over time, this mechanism helps expand access where it is needed most. As of now, this may not be the case: a 2019 report found slightly lower public and private sector facility empanelment in states with higher poverty head count [60].

While analysing the access to specialties across the states, the share of public sector was high in providing care for tertiary care packages. Recent global burden of disease estimates for India reported cardiovascular diseases contribute to 28·1% (95% UI 26·5–29·1) of the total deaths and 14·1% (12·9–15·3) of the total DALYs in 2016 [61]. However, cardiology specialty is offered by only 24% total facilities through the scheme in the country. Emergency care services were available in 71.8% of facilities which are vital in managing road traffic injuries which are among the top 15 causes of mortality in the country [62]. Another leading cause for mortality is chronic kidney disease and despite dialysis being reported as the most sought after procedure under PMJAY, however nephrology as specialty package was not reported under the scheme [63]. This service and burden mismatch should be addressed through empanelment as the program advances.

Prior research has shown that eligible households do not access care due to the supply side constraints in the form of fewer hospitals in their vicinity, and that there is a strong negative correlation between state poverty levels and specialty hospital empanelment [60]. As aforementioned gaps in service availability, while associated with portability of care, are paradoxically associated with restricted beneficiary access to local specialty care. Also, it was found that not all private facilities were providing specialties like General Surgery, Obstetrics & Gynaecology, Orthopaedics and Urology, meaning that the comprehensiveness of coverage was inconsistent.

## Limitations

While analysing the data we observed the following data quality issues which may affect the validity of findings. The presence of large number of SHCs and PHCs, some of them reported to be offering even tertiary care services like paediatric cancer management raises serious concerns over quality of data in the central registry. Public facilities often get upgraded to higher level facilities, but their old names may not be upgraded in the website which might be a reason for large number of SHCs and PHCs. The specialty department in medical colleges of Tamil Nadu were counted as separate institutions in the database which amplifies the count empanelled facilities under the scheme. Moreover, population adjusted figures use Census 2011 estimates, population sizes have obviously grown in the years since the last Census and it would be most appropriate to use updated figures to compute more precise coverage estimates. Given the scope of this analysis, we were unable to explore reasons for patterns of empanelment across and within states, address the impact of empanelment on population, service and system outcomes, or reflect on aspects like budgetary allocations and claims utilisation across settings. These are clearly critical areas of further inquiry.

## Conclusion

This study undertook to characterise patterns of empanelment under PMJAY nationally. We found that a majority of the hospital empanelled under the scheme are in states with previous experience of implementing publicly funded health insurance schemes, with the exception of Uttar Pradesh. Reasons underlying these patterns of empanelment as well as the impact of empanelment on service access, utilisation, population health and financial risk protection warrant further study. While the inclusion and regulation of the private sector is a goal that

may be served by empanelment, the role of public sector remains critical, particularly in underserved areas of India.

## Supporting information

**S1 Table. Distribution of public sector PMJAY empanelment across levels of care and by mode of implementation.**
(DOCX)

**S2 Table. State wise distribution of public health care facility empanelment under PMJAY.**
(DOCX)

**S3 Table. State wise distribution of public health care facility empanelment under PMJAY by geography.**
(DOCX)

**S4 Table. State wise estimates of number of beds in public empanelled health care facilities under PMJAY.**
(DOCX)

**S5 Table. State wise estimates of number of beds in public empanelled health care facilities under PMJAY by geography.**
(DOCX)

**S6 Table. State wise availability of public empanelled hospitals per hundred thousand eligible population by mode of implementation.**
(DOCX)

**S1 File.**
(XLSX)

## Acknowledgments

We are grateful to Dr. Shalini Singh, Associate Faculty, Health Systems, Department of International Health, John Hopkins Bloomberg School of Public Health for her key reflections and critical inputs. We also acknowledge our collaboration with ITAD Ltd.

## Author Contributions

**Conceptualization:** Jaison Joseph.

**Data curation:** Jaison Joseph, Hari Sankar D.

**Formal analysis:** Jaison Joseph.

**Funding acquisition:** Devaki Nambiar.

**Methodology:** Jaison Joseph, Hari Sankar D., Devaki Nambiar.

**Resources:** Devaki Nambiar.

**Supervision:** Devaki Nambiar.

**Validation:** Hari Sankar D., Devaki Nambiar.

**Writing – original draft:** Jaison Joseph.

**Writing – review & editing:** Jaison Joseph, Hari Sankar D., Devaki Nambiar.

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
