## [Decision Letter · Decision Letter 0]

11 Jan 2021

PONE-D-20-33726

Empanelment of Health Care Facilities under Ayushman Bharat Pradhan Mantri Jan Arogya Yojana (AB PM-JAY) in India

PLOS ONE

Dear Dr. Joseph,

Thank you for submitting your manuscript to PLOS ONE. After careful consideration, we feel that it has merit but does not fully meet PLOS ONE’s publication criteria as it currently stands. Therefore, we invite you to submit a revised version of the manuscript that addresses the points raised during the review process.

All four reviewers are favourable for recommending this paper with a minor revision. However, my observation on this paper is it needs a major revision. So, I go with a decision of major revision.  I have enclosed my comments for authors as below.

The paper currently looks more of a descriptive in nature. Lacks analytical rigour and critical/logical interpretations. Some of the interpretations of the findings look indolent and show a sloppiness.The conclusion doesn’t support with data. Authors need a little more intriguing interpretations and conclusions. For instance, the authors write: “the current distribution of empanelled AB-PMJAY hospitals favours better performing states”. In what terms authors saying better performing states. If I look at the five states that authors listing on socio-economic and health status grounds, only two states are better performing states. Rajasthan, Uttar Pradesh and Gujarat are poor and moderately performing in terms of health care and outcome indicators. The possible reason for the larger contribution from these states needs to be explained. Maybe one reason could be all of them are BJP ruling states or its alliance ruling states at the time of programme conceptualisation and initiation. So, the state role needs to be clearly drawn in this case.Also, the authors must explain the reasons why Private Sectors hospitals not actively taking part in the programme from several angles, especially from the side of state subsidies vis-a-vis the regular out-pocket expenses of hospitals in this sector.  Thus, how to reach a consensus on these grounds?Some states following a much better health care schemes than AB-PMJAY for instance, Telangana and Andhra Pradesh. Thus, the states private hospitals have mostly partnered with states programmes than central programmes. Some of these intricacies missing from the paper.Also, authors critically need to evaluate how far private insurance-based health care schemes work in a highly privatised, compartmentalised and hierarchical health care service delivery system like India.  The paper completely silent on budgetary allocations to meet the stated goals.

We look forward to receiving your revised manuscript.

Kind regards,

Srinivas Goli, Ph.D.

Academic Editor

PLOS ONE

Additional Editor Comments:

All four reviewers are favourable for recommending this paper with a minor revision. However, my observation on this paper is it needs a major revision. So, I go with a decision of major revision. I have enclosed my comments for authors as below.

1. The paper currently looks more of a descriptive in nature. Lacks analytical rigour and critical/logical interpretations. Some of the interpretations of the findings look indolent and show a sloppiness.

2. The conclusion doesn’t support with data. Authors need a little more intriguing interpretations and conclusions. For instance, the authors write: “the current distribution of empanelled AB-PMJAY hospitals favours better performing states”. In what terms authors saying better performing states. If I look at the five states that authors listing on socio-economic and health status grounds, only two states are better performing states. Rajasthan, Uttar Pradesh and Gujarat are poor and moderately performing in terms of health care and outcome indicators. The possible reason for the larger contribution from these states needs to be explained. Maybe one reason could be all of them are BJP ruling states or its alliance ruling states at the time of programme conceptualisation and initiation. So, the state role needs to be clearly drawn in this case.

3. Also, the authors must explain the reasons why Private Sectors hospitals not actively taking part in the programme from several angles, especially from the side of state subsidies vis-a-vis the regular out-pocket expenses of hospitals in this sector. Thus, how to reach a consensus on these grounds?

4. Some states following a much better health care schemes than AB-PMJAY for instance, Telangana and Andhra Pradesh. Thus, the states private hospitals have mostly partnered with states programmes than central programmes. Some of these intricacies missing from the paper.

5. Also, authors critically need to evaluate how far private insurance-based health care schemes work in a highly privatised, compartmentalised and hierarchical health care service delivery system like India.

6. The paper completely silent on budgetary allocations to meet the stated goals.

"We are in full adherence of Plos One standard on sharing data and materials. I wish to confirm that all authors have no competing interests to declare."

We note that one or more of the authors are employed by a commercial company: ACCESS Health International Inc.

(2) Please also provide an updated Competing Interests Statement declaring this commercial affiliation along with any other relevant declarations relating to employment, consultancy, patents, products in development, or marketed products, etc.  

3. Please upload a copy of Figure 4, to which you refer in your text on page 20. If the figure is no longer to be included as part of the submission please remove all reference to it within the text.

Reviewers' comments:

Reviewer's Responses to Questions

**Comments to the Author**

1. Is the manuscript technically sound, and do the data support the conclusions?

Reviewer #1: No

Reviewer #2: Yes

Reviewer #3: Yes

Reviewer #4: Yes

2. Has the statistical analysis been performed appropriately and rigorously? 

Reviewer #1: No

Reviewer #2: Yes

Reviewer #3: I Don't Know

Reviewer #4: Yes

3. Have the authors made all data underlying the findings in their manuscript fully available?

Reviewer #1: Yes

Reviewer #2: Yes

Reviewer #3: No

Reviewer #4: No

4. Is the manuscript presented in an intelligible fashion and written in standard English?

Reviewer #1: No

Reviewer #2: Yes

Reviewer #3: Yes

Reviewer #4: Yes

5. Review Comments to the Author

Reviewer #1: 1.The objective of study is not stated and data presented its analysis in the study does not support the conclusion.

2.A large section of introduction and discussion appear to copy and pasted.

3.The entire introduction section needs to rewritten which focus on what the author intent to achieve though the study.

4.The result do not support the conclusion

5.The discussion section of the study is vague and does not support the conclusion.

Reviewer #2: Indeed a good attempt to do the secondary data analysis and given a good insight to the PMJAY scheme in India.

Narration was good. Very well explained with necessary tables and figures.

Please be uniform while converting INR to USD all through the manuscript (Line no. 91)

Considering India having large geographical area, result could have discussed in terms of regions (South, North, Central, Eastern) rather than individual states.

Reviewer #3: I believe the article is much relevant to scientific literature concerning the access to healthcare services under PM-JAY. It is interesting to find that public sector dominates empanelment across states in India. I enjoyed reading the article and how it concludes saying that although there is a need to regulate and include more private hospitals, it is also critical to ensure and maintain the role of public sector in underserved areas.

Additional comments:

• Abstract - conclusion section: In Line 46 the authors mention “better performing states”. I would like to see it written specifically with regard to what aspects they are better performing.

Line 49- the authors mention that finding the appropriate balance of in purchase of care is critical to the success of any publicly financed insurance scheme in India. How do we measure success? I think the authors could provide the specific detail of success of PM-JAY in terms of coverage, or effectiveness of the health insurance scheme.

• In the Methods section, the authors need to mention about why the ethics statement was not needed for this study.

• The authors have used the term cross sectional in line 32 and line 168. This is not a cross sectional study design in my understanding since it doesn’t involve primary research. The study methodology could be written as “secondary analysis of cross-sectional data using descriptive statistical methods”.

• Table 1. Key Features of the Pradhan Mantri Jan Arogya Yojana (PMJAY),

Under Who uses the services? Section, Point “a” mentions “Enrolled Population falling under the category” is not clear to me.

Does the word “category” in point “a” refer to the ones mentioned in b, c and d? I would like to see if that part can be clarified.

• Line 177: citation 34 – the link directs to a page with the message “forbidden access”.

• Line 180: The Database that has been developed as mentioned and link given on reference number 35 is not accessible. I would recommend the authors to check the submission guidelines so as to meet the publication criteria with regard to fulfilling the criteria for validation, utilization and availability of database. “The Database should ideally discuss plans for long term database growth, maintenance and stability. Authors should provide a direct link to the database hosting site from within the paper.” The link is provided in the manuscript in reference number 35 but I have not been able to access the same.

• Line 234: In my understanding, when the term “only” is used, the authors seem to be inclined towards the empanelment of more private facilities which may or may not guarantee effectiveness of PM JAY coverage as per the studies that I have come across. Or if the authors prefer to stick to that statement then they may choose to highlight more studies that show why empanelment of private facilities has shown to be effective or also provide evidence as to why creating a balance between private and public is crucial. Although they have addressed the same, I think it is not enough. It would be good to ponder over the following questions. 1)Do we solely take the patient choice of providers/facility into consideration 2) Do we consider how effective the coverage can be regardless of the type of facility (public/private) especially in terms of reduction in the out-of-pocket expenditure and catastrophic expenditure for the enrolled patients. 3) Would empanelment of private/public facilities improve access or financial protection for the enrolled patients? The same goes for the statement in line 412 where the authors mention about the over representation of public facilities. I hope the authors understand what I am trying to say. If they would like additional insights, they may refer to literature on the suitability of publicly funded purchasing from private providers in the Indian Context.

• Line 336 and 340: I am unable to read the articles as the link directs to a website that mentions error/ page not found. (Citation number 39 and 40)

• I am in total agreement with Line 352 where the authors mention that Participation of the private sector in public health insurance is a critical area for further research and policy attention.

Reviewer #4: This paper presents an analysis of public data from the PH-JAY program, which is the world’s largest health assurance scheme providing health cover to about 500 million of the poorest Indians. The issue addressed by the paper is undoubtedly both an important and pertinent one, and has far-reaching implications for the many millions of vulnerable people served by the program.

Regarding rigour and reporting: the underlying logic and philosophy of the work is made clear, the objectives outlined well, and the approach and methods used to address them are presented well. The findings of this study offer very important lessons for improving the AB PM-JAY, and these could be communicated more clearly accessible to the reader to improve the impact of this paper. To this end, I offer the following recommendations:

Abstract

- The states named in the Results should be placed in order of the percentage of empanelled PMJAY facilities.

- The statements made in the conclusion are somewhat ambiguous. The authors could clarify what they mean by the “distribution of empanelled AB-PMJAY hospitals favours better performing states”. Firstly, does ‘better performing” allude to the state’s population health status, or the state’s health system, or something else? It is unclear what is meant by ‘favours’. It might be easier to simply state, for example, that most of the empanelled hospitals are located within states that already have well-functioning health system performance.

- The implications of this finding for policymakers could easily be outlined in the final sentence; i.e. highlight the need for greater empanelment of hospitals in states where it is most needed. In its current form, the final sentence is somewhat ambivalent in its tone and could do with some rewriting.

- Page 3 line 48-49: “appropriate balance of in purchase of care”. This sentence should be edited to use either ‘of’ or ‘in’.

Introduction

- The authors do well to introduce the key concepts and ground them within the international development agenda of the United Nations. The ongoing operationalization of PM-JAY is discussed in the context of cooperative federalism in the Indian union, and the contrasts in healthcare equity (e.g. RSBY) between urban and rural India.

- Page 5, Line 90 - 91: what is ‘cashless mode’ and a ‘floater basis’?

- Page 6, line 113: add a hyphen ‘urban-centric’

- Spelling and grammar:

o Page 5, line 81:“…financial protection for secondary and tertiary care to about bottom 40% of…” Please remove ‘about’.

Methods:

- These have been well described in sufficient detail to facilitate replication.

- The public data that was used in the analysis should be made available in a suitably formatted file (and a link provided in the manuscript), in accordance with the PLOS Data policy.

Results

- A lot of data has been presented here, including 7 tables. While the well-illustrated figures certainly make the findings more accessible, the sheer number of large tables do appear to compromise the readability of the manuscript, and thus risk obscuring the important messages buried therein.

- I would recommend that the authors consider placing large unwieldy tables in the supplementary material while making reference to them and their findings in the main body of the manuscript. Additionally, it is recommended that the authors avoid presenting the same data in both the table and figures. Wherever data has been presented in a figure with data values labelled, it is no longer necessary to also present this in a table.

Discussion

- The discussion identifies several issues of high importance, for example, the primary care coverage, lack of cardiology and nephrology services, etc. I request that the authors compile and present a discrete, bulleted list of policy recommendations based on the critical analysis outlined in the discussion. Not only would this improve the accessibility of the study’s findings and highlight the importance of this analysis, but this would go some way towards fulfilling the social responsibility of research such as this: to actually improve human health. The recommendations could be presented in a box or figure for better readability and could also be outlined in the final part of the abstract. Such presentation would provide ideal messages for effective media communication in the post-publication period.

6. PLOS authors have the option to publish the peer review history of their article (what does this mean?). If published, this will include your full peer review and any attached files.

Reviewer #1: **Yes: **Dr Nishant Kumar

Reviewer #2: **Yes: **Dr. Ramesh Holla

Reviewer #3: No

Reviewer #4: No

---

## [Author Response · Author response to Decision Letter 0]

26 Mar 2021

Empanelment of Health Care Facilities under Ayushman Bharat Pradhan Mantri Jan Arogya Yojana (AB PM-JAY) in India; PONE-D-20-33726

Response for Additional Editor Comments

1. The paper currently looks more of a descriptive in nature. Lacks analytical rigour and critical/logical interpretations. Some of the interpretations of the findings look indolent and show a sloppiness.

Thank you for the critical feedback. This analysis is based on a data of empanelled health facilities, therefore only basic descriptive inferences were made. We don’t intend for the results to look sloppy – while additional analyses would be desirable, we feel there are already novel contributions based on this work, worthy of publishing. We are encouraged that other reviewers have not rejected our argument, but rather offered critical revisions to improve it.

2. The conclusion doesn’t support with data. Authors need a little more intriguing interpretations and conclusions. For instance, the authors write: “the current distribution of empanelled AB-PMJAY hospitals favours better performing states”. In what terms authors saying better performing states. If I look at the five states that authors listing on socio-economic and health status grounds, only two states are better performing states. Rajasthan, Uttar Pradesh and Gujarat are poor and moderately performing in terms of health care and outcome indicators. The possible reason for the larger contribution from these states needs to be explained. Maybe one reason could be all of them are BJP ruling states or its alliance ruling states at the time of programme conceptualisation and initiation. So, the state role needs to be clearly drawn in this case. 

We agree that our findings are not dramatic and indeed that empanelment is not greater only in better performing states. However, there are an important foundation on which to advance other areas of inquiry. The “why” of these findings is out of scope of the current analysis and would make use of methods – political economy analysis etc. We make mention of this in our discussion on page 23. 

The editor may appreciate that we as authors are not looking at the “why” question using a purely academic, political economy lens as that would affect our ability to work in and partner with states, precisely for the reasons of political economy that can be studied or explored at a distance, but come at a cost in terms of applied health systems research.

3. Also, the authors must explain the reasons why Private Sectors hospitals not actively taking part in the programme from several angles, especially from the side of state subsidies vis-a-vis the regular out-pocket expenses of hospitals in this sector. Thus, how to reach a consensus on these grounds?

The data we accessed for the study will not provide answers to the question of why private hospitals are actively taking part, as we are looking only at the empanelment and facility distribution provided by NHA. We tried to examine the literature on this but were not able to find any published evidence. However, we did find evidence on private sector participation in publicly funded health insurance in relation to poverty status of the state and have referenced this in the discussion, on page 18. Further research is required to answer the above questions which we have also mentioned on our limitation section on page 28. 

4. Some states following a much better health care schemes than AB-PMJAY for instance, Telangana and Andhra Pradesh. Thus, the state’s private hospitals have mostly partnered with states programmes than central programmes. Some of these intricacies missing from the paper. 

In the current study, our aim is to describe the pattern and distribution of hospitals under AB-PMJAY. We looked for published literature that describes the partnership models of certain states in the insurance space and were not able to find such resources. We’d be grateful if the editor could point us to towards such resources so we may add them in our discussion. 

5. Also, authors critically need to evaluate how far private insurance-based health care schemes work in a highly privatised, compartmentalised and hierarchical health care service delivery system like India. 

We agree that the landscape in India is highly privatised; yet the relationship of this to empanelment is vexed, as we point out on page 8 (i.e. empanelment of the private sector is low in states like Bihar and Chhattisgarh). Beyond this, a critical evaluation of how private schemes work is not in the scope of our study as we were only focussing on the distribution of public/private facilities under PMJAY scheme using the empanelment data. 

6. The paper completely silent on budgetary allocations to meet the stated goals. 

This is an absolutely critical analysis that we have recommended in our limitations of study on page 28. However, our focus is on the empanelment issue beyond which we are not clear on what budgetary allocations the editor is referring to. Do you mean overall funding for PMJAY? Or is this in reference to claims processing? This data is often not easily available and while important – we have tried to explore issues we feel are relevant on page 29. 

Response to Reviewers Comments to the Author

Reviewer #1: 

1. The objective of study is not stated, and data presented its analysis in the study does not support the conclusion.

Thank you for the comment, objectives of our study are stated at the end of the introduction section, on page 10-11. We have, moreover rephrased and re-written sections for clearer readability and linkage across results and conclusions (see pages 13-23).

2. A large section of introduction and discussion appear to copy and pasted.

We were surprised to see this comment! We did a plagiarism check and, the percentage of repeated text is less than 10%. However, based on this comment, we have revised the introduction (5-11) and discussion sections (23-28).

3.The entire introduction section needs to re-written which focus on what the author intent to achieve though the study.

Thank you for the comment; our introduction (5-11) has been revised focusing on the aims and objectives of the study. 

4.The result do not support the conclusion. The discussion section of the study is vague and does not support the conclusion.

This point is well taken. We have revised the conclusion (page 29) substantially to reflect the results (12-33) and also made appropriate changes to the discussion section (23-22) accordingly.

Reviewer #2: 

1. Please be uniform while converting INR to USD all through the manuscript (Line no. 91)

Suggestion well taken and changes made where relevant.

2. Considering India having large geographical area, result could have discussed in terms of regions (South, North, Central, Eastern) rather than individual states.

Considering the suggestions, we have added the major tables of the study geographically and the same is added as Supplementary File (S1 File).

Reviewer #3: 

1. Abstract - conclusion section: In Line 46 the authors mention “better performing states”. I would like to see it written specifically with regard to what aspects they are better performing.

Thank you for the comment, we have removed the statement better performing state, changes have been made in page 4 of abstract conclusion section

2. Line 49- the authors mention that finding the appropriate balance of in purchase of care is critical to the success of any publicly financed insurance scheme in India. How do we measure success? I think the authors could provide the specific detail of success of PM-JAY in terms of coverage, or effectiveness of the health insurance scheme.

It is true that measuring success has many dimensions in the case of insurance schemes. PMJAY has shown limitations and strengths in coverage and the evidence on effectiveness is limited as of now. We point this out in our discussion on pages 23-28.

3. In the Methods section, the authors need to mention about why the ethics statement was not needed for this study.

This was an inadvertent omission. An ethics statement is now included in the methods on page 11.

4. The authors have used the term cross sectional in line 32 and line 168. This is not a cross sectional study design in my understanding since it doesn’t involve primary research. The study methodology could be written as “secondary analysis of cross-sectional data using descriptive statistical methods”.

The suggestion is well taken, the changes have been made in the abstract page 3 as well as the methods page 11

5. Table 1. Key Features of the Pradhan Mantri Jan Arogya Yojana (PMJAY),

Under Who uses the services? Section, Point “a” mentions “Enrolled Population falling under the category” is not clear to me. Does the word “category” in point “a” refer to the ones mentioned in b, c and d? I would like to see if that part can be clarified.

The suggestion is well taken, the changes have been made in Table 1 under “Who uses the services?”

6. Line 177: citation 34 – the link directs to a page with the message “forbidden access”.

Thank you for pointing out the error. This issue happened due to site upgradation by PMJAY post data collection. The chance of recurrence of this issue cannot be avoided as there may be changes in the data URL with every website upgradation.

7. Line 180: The Database that has been developed as mentioned and link given on reference number 35 is not accessible. I would recommend the authors to check the submission guidelines so as to meet the publication criteria with regard to fulfilling the criteria for validation, utilization and availability of database. “The Database should ideally discuss plans for long term database growth, maintenance and stability. Authors should provide a direct link to the database hosting site from within the paper.” The link is provided in the manuscript in reference number 35 but I have not been able to access the same.

The dataset used for this study is now submitted as Supplemental File (S 2 File). 

8. Line 234: In my understanding, when the term “only” is used, the authors seem to be inclined towards the empanelment of more private facilities which may or may not guarantee effectiveness of PM JAY coverage as per the studies that I have come across. Or if the authors prefer to stick to that statement then they may choose to highlight more studies that show why empanelment of private facilities has shown to be effective or also provide evidence as to why creating a balance between private and public is crucial. Although they have addressed the same, I think it is not enough. It would be good to ponder over the following questions. 1)Do we solely take the patient choice of providers/facility into consideration 2) Do we consider how effective the coverage can be regardless of the type of facility (public/private) especially in terms of reduction in the out-of-pocket expenditure and catastrophic expenditure for the enrolled patients. 3) Would empanelment of private/public facilities improve access or financial protection for the enrolled patients? The same goes for the statement in line 412 where the authors mention about the over representation of public facilities. I hope the authors understand what I am trying to say. If they would like additional insights, they may refer to literature on the suitability of publicly funded purchasing from private providers in the Indian Context.

The reviewer brings up critical points. We had referenced the literature on publicly funded health insurance as part of our discussion on page 24. Some of these questions cannot be answered from our analysis, which we indicate in our limitations on page 28 and recommendations for further analysis across the discussion. 

9. Line 336 and 340: I am unable to read the articles as the link directs to a website that mentions error/ page not found. (Citation number 39 and 40)

Thank you for pointing this out, the citation has been updated 

10. I am in total agreement with Line 352 where the authors mention that Participation of the private sector in public health insurance is a critical area for further research and policy attention.

Thank you for your comment 

Reviewer #4: 

This paper presents an analysis of public data from the PH-JAY program, which is the world’s largest health assurance scheme providing health cover to about 500 million of the poorest Indians. The issue addressed by the paper is undoubtedly both an important and pertinent one, and has far-reaching implications for the many millions of vulnerable people served by the program.

Regarding rigour and reporting: the underlying logic and philosophy of the work is made clear, the objectives outlined well, and the approach and methods used to address them are presented well. The findings of this study offer very important lessons for improving the AB PM-JAY, and these could be communicated more clearly accessible to the reader to improve the impact of this paper. To this end, I offer the following recommendations:

1. Abstract - The states named in the Results should be placed in order of the percentage of empanelled PMJAY facilities.

We thank the reviewer for the suggestion, changes are made on page 3

2. The statements made in the conclusion are somewhat ambiguous. The authors could clarify what they mean by the “distribution of empanelled AB-PMJAY hospitals favours better performing states”. Firstly, does ‘better performing” allude to the state’s population health status, or the state’s health system, or something else? It is unclear what is meant by ‘favours’. It might be easier to simply state, for example, that most of the empanelled hospitals are located within states that already have well-functioning health system performance.

Suggestion is well taken, although as other reviewers have pointed out, not all the states with high empanelment have well-functioning health system. We have revised the text to reflect this on page 3-4

3. The implications of this finding for policymakers could easily be outlined in the final sentence; i.e. highlight the need for greater empanelment of hospitals in states where it is most needed. In its current form, the final sentence is somewhat ambivalent in its tone and could do with some rewriting.

Thank you for the comment, the section has been re-written in both the abstract and conclusion page 3-4

4. Page 3 line 48-49: “appropriate balance of in purchase of care”. This sentence should be edited to use either ‘of’ or ‘in’.

Suggestion is well taken, and changes on pages 3-4

Introduction - The authors do well to introduce the key concepts and ground them within the international development agenda of the United Nations. The ongoing operationalization of PM-JAY is discussed in the context of cooperative federalism in the Indian union, and the contrasts in healthcare equity (e.g. RSBY) between urban and rural India.

5. Page 5, Line 90 - 91: what is ‘cashless mode’ and a ‘floater basis’?

Suggestion well taken, explanation provided for cashless mode and floater basis in page 7

6. Page 6, line 113: add a hyphen ‘urban-centric’

Suggestion well taken; changes made in page 9

7. Spelling and grammar: Page 5, line 81: “…financial protection for secondary and tertiary care to about bottom 40% of…” Please remove ‘about’.

Suggestion well taken; changes made in page 6

8. Methods:

- These have been well described in sufficient detail to facilitate replication.

- The public data that was used in the analysis should be made available in a suitably formatted file (and a link provided in the manuscript), in accordance with the PLOS Data policy.

Thank you for the suggestion, all the data that were for the analysis shall be made available publicly in compliance with PLOS data policy as Supplementary File S 2 File

9. Results

- A lot of data has been presented here, including 7 tables. While the well-illustrated figures certainly make the findings more accessible, the sheer number of large tables do appear to compromise the readability of the manuscript, and thus risk obscuring the important messages buried therein.

Thank you for the suggestion, Many of the tables are now incorporated in Supplementary file S1 File

- I would recommend that the authors consider placing large unwieldy tables in the supplementary material while making reference to them and their findings in the main body of the manuscript. Additionally, it is recommended that the authors avoid presenting the same data in both the table and figures. Wherever data has been presented in a figure with data values labelled, it is no longer necessary to also present this in a table.

Thank you for the suggestion, we have moved the large table to Supplementary file S1File 

9. Discussion

- The discussion identifies several issues of high importance, for example, the primary care coverage, lack of cardiology and nephrology services, etc. I request that the authors compile and present a discrete, bulleted list of policy recommendations based on the critical analysis outlined in the discussion. Not only would this improve the accessibility of the study’s findings and highlight the importance of this analysis, but this would go some way towards fulfilling the social responsibility of research such as this: to actually improve human health. The recommendations could be presented in a box or figure for better readability and could also be outlined in the final part of the abstract. Such presentation would provide ideal messages for effective media communication in the post-publication period.

A number of suggestions have emerged from this research, particularly for more research(!). We have indicated these in the abstract as well as the discussion section (pages 3-4 & 23). Listing them in a box makes them somewhat disconnected from context, so we haven’t done this, but if the reviewer is strongly keen on this, we could try.

We have made the necessary changes to adhere to PLOS One guidelines 

"We are in full adherence of Plos One standard on sharing data and materials. I wish to confirm that all authors have no competing interests to declare."

We note that one or more of the authors are employed by a commercial company: ACCESS Health International Inc.

 The co-author from ACESSS have decided to withdraw his name from the paper. so we believe the issue is resolved 

The co-author from ACCESS have decided to withdraw his name from the paper.so we believe the issue is resolved 

The co-author from ACCESS have decided to withdraw his name from the paper so we believe the issue is resolved 

(2) Please also provide an updated Competing Interests Statement declaring this commercial affiliation along with any other relevant declarations relating to employment, consultancy, patents, products in development, or marketed products, etc. 

The co-author from ACCESS have decided to withdraw his name from the paper so we believe the issue is resolved 

2. Please upload a copy of Figure 4, to which you refer in your text on page 20. If the figure is no longer to be included as part of the submission please remove all reference to it within the text.

This was a manual error which have been resolved by making in the correction in the manuscript.

---

## [Decision Letter · Decision Letter 1]

4 May 2021

Empanelment of Health Care Facilities under Ayushman Bharat Pradhan Mantri Jan Arogya Yojana (AB PM-JAY) in India

PONE-D-20-33726R1

Dear Dr. Joseph,

We’re pleased to inform you that your manuscript has been judged scientifically suitable for publication and will be formally accepted for publication once it meets all outstanding technical requirements.

Kind regards,

Srinivas Goli, Ph.D.

Academic Editor

PLOS ONE

Additional Editor Comments (optional):

Considering reviewers opinion and my own reading of this paper, I am recommending it for publication.

Reviewers' comments:

Reviewer's Responses to Questions

**Comments to the Author**

1. If the authors have adequately addressed your comments raised in a previous round of review and you feel that this manuscript is now acceptable for publication, you may indicate that here to bypass the “Comments to the Author” section, enter your conflict of interest statement in the “Confidential to Editor” section, and submit your "Accept" recommendation.

Reviewer #2: All comments have been addressed

Reviewer #4: All comments have been addressed

2. Is the manuscript technically sound, and do the data support the conclusions?

Reviewer #2: Yes

Reviewer #4: Yes

3. Has the statistical analysis been performed appropriately and rigorously? 

Reviewer #2: Yes

Reviewer #4: I Don't Know

4. Have the authors made all data underlying the findings in their manuscript fully available?

Reviewer #2: Yes

Reviewer #4: Yes

5. Is the manuscript presented in an intelligible fashion and written in standard English?

Reviewer #2: Yes

Reviewer #4: Yes

6. Review Comments to the Author

Reviewer #2: The authors have addressed all the comments raised by the reviewer in the satisfactory manner and all the supporting data was shared along with the manuscript itself.

Reviewer #4: (No Response)

7. PLOS authors have the option to publish the peer review history of their article (what does this mean?). If published, this will include your full peer review and any attached files.

Reviewer #2: **Yes: **Dr. RAMESH HOLLA

Reviewer #4: **Yes: **Dr. Myron Anthony Godinho

---

## [Editor Report · Acceptance letter]

17 May 2021

PONE-D-20-33726R1 

Empanelment of Health Care Facilities under Ayushman Bharat Pradhan Mantri Jan Arogya Yojana (AB PM-JAY) in India 

Dear Dr. Joseph:

I'm pleased to inform you that your manuscript has been deemed suitable for publication in PLOS ONE. Congratulations! Your manuscript is now with our production department. 

Kind regards, 

on behalf of

Dr. Srinivas Goli 

Academic Editor

PLOS ONE